# Calcium stabilizes the strongest protein fold

Lukas F. Milles [1], Eduard M. Unterauer[1], Thomas Nicolaus[1] & Hermann E. Gaub[1]

*Staphylococcal* pathogens adhere to their human targets with exceptional resilience to mechanical stress, some propagating force to the bacterium via small, Ig-like folds called B domains. We examine the mechanical stability of these folds using atomic force microscopy-based single-molecule force spectroscopy. The force required to unfold a single B domain is larger than 2 nN – the highest mechanostability of a protein to date by a large margin. B domains coordinate three calcium ions, which we identify as crucial for their extreme mechanical strength. When calcium is removed through chelation, unfolding forces drop by a factor of four. Through systematic mutations in the calcium coordination sites we can tune the unfolding forces from over 2 nN to 0.15 nN, and dissect the contribution of each ion to B domain mechanostability. Their extraordinary strength, rapid refolding and calcium-tunable force response make B domains interesting protein design targets.

[1] Lehrstuhl für Angewandte Physik and Center for Nanoscience, Ludwig-Maximilians-University, Amalienstr. 54, 80799 Munich, Germany. Correspondence and requests for materials should be addressed to L.F.M. (email: lukas.milles@physik.uni-muenchen.de) or to H.E.G. (email: gaub@lmu.de)

Pathogenic bacteria have evolved to strongly and persistently adhere to their hosts using a variety of mechanisms. Among them are microbial surface components recognizing adhesive matrix molecules (MSCRAMMs), which adhere to target proteins of their human hosts[1,2]. Covalently bound to the Gram-positive peptidoglycan and extruding into the extracellular space, these adhesins target sequences on the order of 15 amino acids in human proteins, notably all chains of fibrinogen, but also other members of the extracellular and adhesive matrix, such as fibronectin[3] and keratin[4]. Initial adhesion is crucial to begin infection, especially under hydrodynamic forces. The ligand binding region A at the N-terminus of these adhesins employs the "Dock, Lock and Latch" (DLL) mechanism[5,6]. In DLL, the peptide target is tightly confined between the N2 and N3 domain of region A, achieving mechanically hyperstable adhesion to host proteins. These adhesins, such as SdrG from *S. epidermidis*, and its homologs from *S. aureus* (ClfA, ClfB, Bbp, FnBPA, and SdrE) can withstand forces in the 2 nN force regime[7–12], approaching the strength of a covalent bond[13]. This extreme strength is achieved through a recently identified molecular mechanism[9]. Along their stalks are so-called B domains, which appear in adhesins such as *S.aureus* SdrD and Bbp, *S. epidermidis* SdrG, and *S. saprophyticus* UafA[14]. B domains are small domains (~ 13 kDa) that link the ligand binding region to a sortase motif, which mediates covalent anchoring of the adhesin to the bacterial peptidoglycan[15]. Thus, it is to be expected that these folds must propagate the extreme mechanical force withstood by the DLL adhesin.

Using atomic force microscopy-based (AFM) single-molecule force spectroscopy (SMFS)[16–19] we investigate the mechanical strength of B domains from *S. epidermidis* SdrG[20] and *S. aureus* SdrD[21,22]. We found their mechanostability to be exceptional, far exceeding all other proteins investigated to date. B domains unfold at forces larger than 2 nN – a strength reminiscent of breaking a covalent bond. In comparison, pili domains of FimA from *Actinomyces oris* have been shown to unfold at ~ 0.7 nN[23] and cohesin domains from cellulosomal bacteria unfold at ~ 0.6 nN[24,25], both at similar force loading rates. Through site-directed mutations, we demonstrate that this stability rests on the coordination of calcium ($Ca^{2+}$) ions. Each B domain coordinates three $Ca^{2+}$ ions in different positions. When these are chelated from the domain their mechanostability drastically decreases by a factor of four – yet forces are still in the vicinity of 0.6–0.8 nN. Systematically incapacitating the $Ca^{2+}$ coordination sites revealed which $Ca^{2+}$ ion is most important to the mechanostability. Furthermore, there are subtle differences between B domains from related organisms – even the same gene – opening multiple scenarios for their role in pathogen adhesion.

## Results

### The SdrG B1 domain unfolds at forces over 2 nN.
The B domains from staphylococcal adhesin SdrG, B1, and B2, act as a linker between the N-terminal A region, where domains N2 and N3 bind host targets with extremely high resilience to mechanical force (Fig. 1a), and the C-terminus, which is covalently anchored to the bacterial peptidoglycan via a sortase motif (Figs. 1b, c). Thus, the B domains are located between an extremely mechanostable non-covalent interaction and a covalent bond, motivating our investigation into their force resilience. Initial experiments probed the unfolding forces of the SdrG B1 domain using its wild type (WT), adjacent ultrastable protein handle, the N2 and N3 domain (Fig. 1d). SdrG N2N3 binding a 15 amino-acid peptide from the N-terminus of fibrinogen ß (Fgß) withstands >2 nN in force. Alternatively, we used the clumping factor B N2 and N3 domains (ClfB) from *S. aureus* as a handle, which

binds a 12 amino-acid C-terminal peptide of dermokine (DK) and conveniently has no B domains, as well as a higher unbinding force than SdrG (Key plasmids for this study were deposited with Addgene and can be found in Table 1). The unfolding forces of SdrG B1 were consistently in the range of 2 nN (Figs. 1e, f). Traces containing no unfolding events before the handle ruptured indicate that sometimes SdrG B1 domain stability even exceeded that of the N2N3 handle. The contour length increment of the unfolding event matched the expected length for an unfolded B1 domain (110 amino acids × 0.365 nm per residue – 4 nm folded protein = 36 nm). Previous cell-based force spectroscopy work on SdrG had described an event preceding complex rupture at comparable forces and extension increments – yet not identified it as a B domain[7]. The SdrG B1 domain was located on the cantilever, whose apex (radius ~ 10 nm) can only present a limited number of molecules. Yet, its unfolding appeared in almost every trace for more than 12 h ($N = 3712$ events), thus we conclude that it refolds. A dynamic force spectrum was acquired for the high-force unfolding of B1 (Fig. 1f), which was described well by the Bell-Evans (BE) model. Curiously, a second unfolding population with an identical contour length increment unfolding ~ 600 pN also emerged (shown in Fig. 2a), hinting at a second, weaker unfolding pathway (excluded from Fig. 1f).

### $Ca^{2+}$-binding sites govern the mechanostability of SdrG B1.
A crystal structure of the SdrG B1 domain is not available to date. Fortunately, a homolog's (SdrD from *Staphylococcus aureus*) B1 domain structure (PDB 4JDZ, alignment see Supplementary Fig. 1) had been determined previously[26]. Homology models of SdrG B1 and B2 could be constructed and were equilibrated using QwikMD configured NAMD molecular dynamics simulation[27–29]. Notably, the B domain adopts an Immunoglobulin (Ig)-like fold containing exclusively ß-strands (Fig. 1c). Furthermore, each B domains coordinates three calcium ions, which were numbered as displayed in Fig. 1b and Fig. 3a. Calcium one to three (Ca1–Ca3) are coordinated mostly via negatively charged side chains. Ca1 is enclosed in a loop, Ca2 lies more solvent exposed and closer to Ca3, which is coordinated by two aspartic acids on the N- and C-terminal ß-strands that close the fold (see Figs. 3a, b).

The presence of calcium is relevant for both folding and stability of many protein domains[30]. Thus, to remove $Ca^{2+}$ ions from the domain, buffers were exchanged introducing a high concentration of the chelating agent ethylenediaminetetraacetic acid (EDTA), which binds divalent ions. When probed in 10 mM EDTA, the stability of SdrG B1 dramatically decreased, and the previously described weak unfolding event ~ 600 pN appeared exclusively (a set of representative force extension curves is shown in Supplementary Fig. 2, for contour length diagram alignments see Supplementary Fig. 3). The SdrG:Fgß interaction remained unaffected by EDTA, despite its $Ca^{2+}$-binding loop. When using citric acid – a more physiological, but milder chelating agent – instead of EDTA, the same weak state emerged (Supplementary Fig. 4), although even at 100 mM citric acid ~ 40% of domains still unfolded from the strong state. As the contour length increment of the weak state remained unchanged compared to the strong unfolding, the B1 domain was still folded in EDTA. The depletion of calcium switched it into a mechanically weaker state, yet it was unclear how many $Ca^{2+}$ were chelated from SdrG B1. The strong state with unfolding events exclusively around 2 nN, was recovered after returning to 10 mM $Ca^{2+}$ (example traces shown in Fig. 2a). This calcium induced stability switching could be repeated for multiple cycles, as shown in Fig. 2b. After EDTA chelation, applying high concentrations of $Mg^{2+}$ did not change SdrG B1 weak state unfolding behavior. Even at 18 mM,

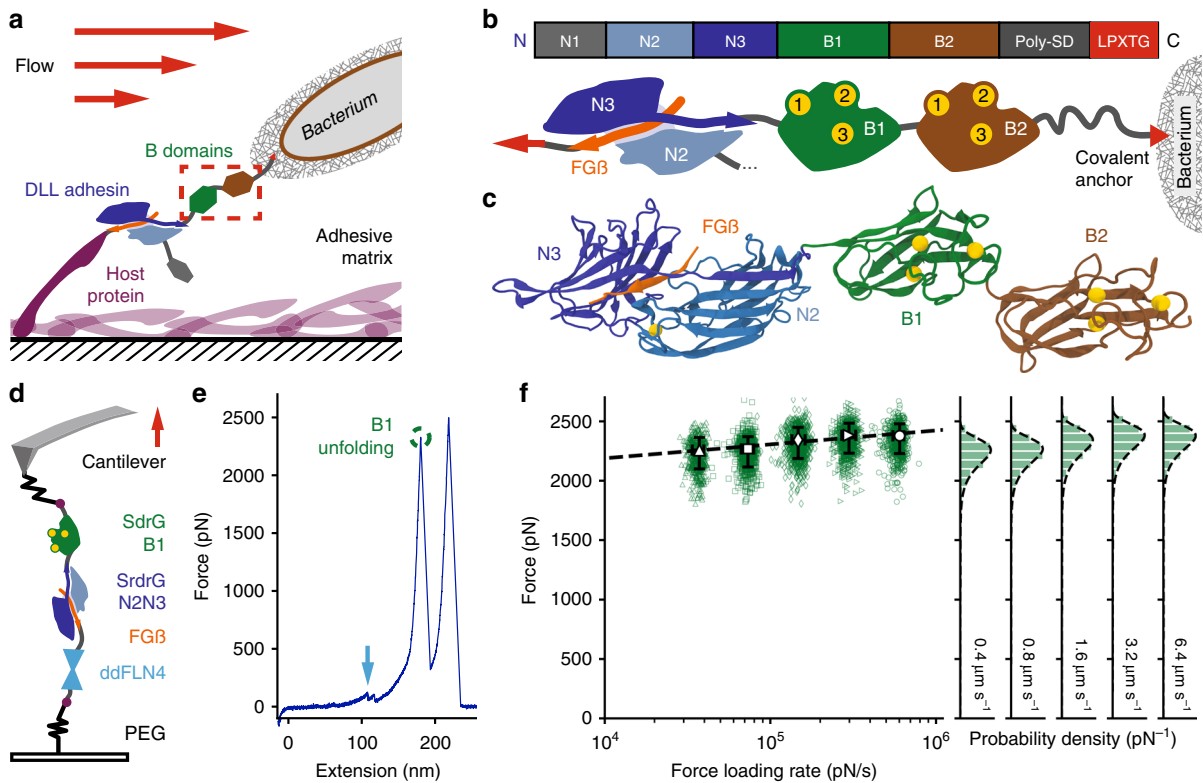

**Fig. 1** Staphylococcal B domains are extremely resilient to mechanical force. **a** B domains, here from SdrG, are the link between the extremely strong interaction between the tip adhesin domain N2N3 (blues) binding a peptide (orange) presented by a host protein covering a surface (in this case fibrinogen, purple). **b** SdrG gene (top) and schematic (bottom): N-terminal N1 domain may be cleaved proteolytically, followed by the N2 and N3 domain that bind Fgß, B1 (green), and B2 (brown) each coordinate three $Ca^{2+}$ ions and connect to the SD repeat region that gives the adhesin its name. The adhesin is covalently anchored to the bacterium's peptidoglycan via a sortase motif (red). **c** MD equilibrated structure of the SdrG N2N3 domains connected to the fully ß-sheet Ig-like folds of the B1 and B2 domain (modeled from the homolog SdrD B1), each B domain coordinates three $Ca^{2+}$ ions (yellow). **d** AFM-SMFS assay: covalent surface anchoring through polyethylene glycol (PEG) via the ybbr-tag (purple) using Fgß (orange)-ddFLN4 (cyan)-ybbr to probe SdrG N2N3-B1-ybbr on the cantilever. **e** Single force-extension trace at 0.8 µm s$^{-1}$ with unfolding of the ddFLN4 fingerprint (cyan arrow) at ~ 100 pN, followed by the SdrG B1 domain (green circle) at >2 nN (with the expected contour length increment of ~ 36 nm). Finally the SdrG N2N3:Fgß complex dissociates, allowing the cantilever to relax to zero force. **f** Dynamic force spectrum for the unfolding of the SdrG B1 domain at retraction velocities: 0.4 µm s$^{-1}$ (triangles, $N = 574$), 0.8 µm s$^{-1}$ (squares, $N = 742$), 1.6 µm s$^{-1}$ (diamonds, $N = 878$), 3.2 µm s$^{-1}$ (forward triangles, $N = 789$), 6.4 µm s$^{-1}$ (circles, $N = 729$). The high N value suggests that the B1 domain refolds on the cantilever. A Bell–Evans model fit (dashed line, $\Delta x = 0.082$ nm, $k_{off}^0 = 3.8E–17$ s$^{-1}$) through the most probable rupture force and force loading rate per velocity (large open markers, errors given as full-width at half maximum for each distribution) confirms the expected log-linear behavior

---

**Table 1 Key plasmids with Addgene accession numbers**

| Plasmid | AddgeneID |
|---|---|
| pET28a-SdrG_N2N3-HIS-ybbr | 101238 |
| pET28a-ClfB_N2N3-HIS-ybbr | 101717 |
| pET28a-SdrG_N2N3-B1-B2-HIS-ybbr | 117979 |
| pET28a-SdrG_N2N3-B1-HIS-ybbr | 117980 |
| pET28a-MGGG-ybbr-HIS-SdrG_B1-DK | 117981 |
| pET28a-MGGG-ybbr-HIS-SdrG_B2-DK | 117982 |
| pET28a-MGGG-ybbr-HIS-SdrD_B1-DK | 117983 |

$Mg^{2+}$ was unable to occupy the $Ca^{2+}$ coordination sites (see Supplementary Fig. 5). The dynamic force spectra for both weak and strong states, shown in Fig. 2c, were determined with a single cantilever. Notably, the dependency of the most probable rupture force on the natural logarithm of the force loading rate in the BE model is almost parallel for both states (strong state: $\Delta x = 0.083$ nm, $k_{off}^0 = 2.8E–17$ s$^{-1}$, weak state $\Delta x = 0.071$ nm, $k_{off}^0 = 0.011$ s$^{-1}$), reflected in similar distances to the transition state $\Delta x$ (within ~ 17% of each other), whereas the large difference in

unfolding force is given through the zero force off-rates $k_{off}^0$, which differ by >14 orders of magnitude.

After they had been exposed to EDTA, inducing the weak state, SdrG B1 domains were returned to $Ca^{2+}$-free buffers (25 mM TRIS, 75 mM NaCl, pH 7.4) and found mostly in the strong state. This puzzling contradiction of $Ca^{2+}$-dependent folding could be explained by trace amounts of contaminating $Ca^{2+}$ in the buffer, estimated to be in the nM range (manufacturer's specifications, see Methods). Thus, the affinity of SdrG B1 for $Ca^{2+}$ must be extremely high. Moving to higher purity $Ca^{2+}$-free reagents (see Methods), a trace $Ca^{2+}$ concentration low enough to keep the domains in the weak state was achieved. Previously, the $Ca^{2+}$-dependent folding and thus $Ca^{2+}$ affinity of homologous B domains from S. aureus SdrD had been measured[21,31]. As our AFM experiments could clearly discern between the strong $Ca^{2+}$ saturated and weak, $Ca^{2+}$-depleted state, we titrated the amount of $Ca^{2+}$ to estimate the affinity of the weak state SdrG B1 domain for $Ca^{2+}$ by plotting the fraction of strong to weak state events against $Ca^{2+}$ concentration. The inflection point was below 1 nM, within our conservative estimate of $Ca^{2+}$ concentration uncertainty, hinting at a sub-nM $K_D$ for $Ca^{2+}$ (Figs. 2d, e).

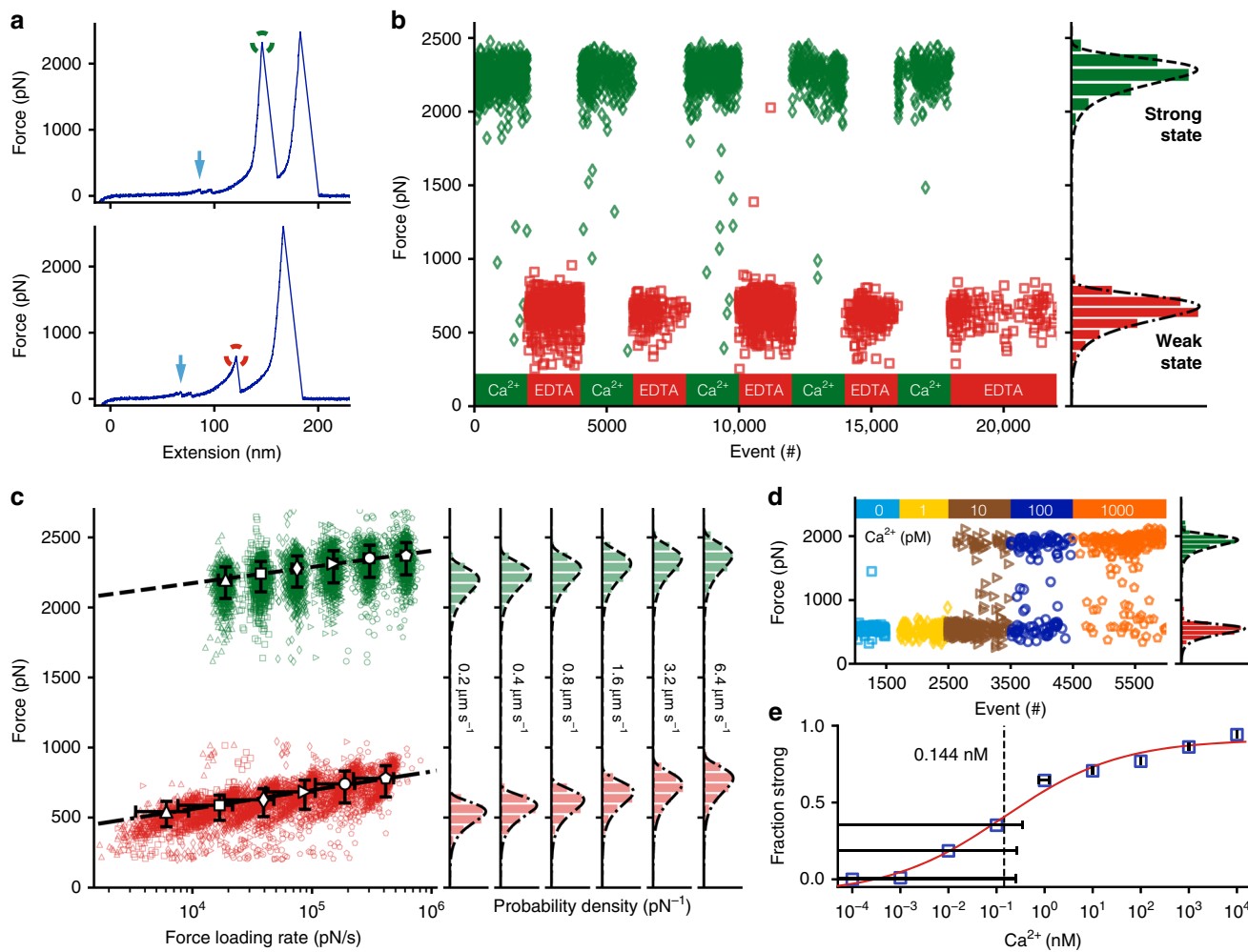

**Fig. 2** B domain stability and unfolding force are governed by calcium. **a** SdrG B1 unfolds in 10 mM Ca$^{2+}$ at over 2 nN (strong state, green circle) and here at ~ 650 pN in the presence of 10 mM EDTA (weak state, red circle) at a retraction velocity of 1.6 μm s$^{-1}$, after ddFLN4 fingerprint unfolding (cyan arrows). **b** SdrG B1 domain stabilities can be cycled repeatedly by alternate application of Ca$^{2+}$ 10 mM (green diamonds) and EDTA 10 mM (red squares). **c** Dynamic force spectrum of the weak and strong state stabilities. Strong state (green) in 10 mM Ca$^{2+}$: 0.2 μm s$^{-1}$ (triangles, $N = 848$), 0.4 μm s$^{-1}$ (squares, $N = 1128$), 0.8 μm s$^{-1}$ (diamonds, $N = 1162$), 1.6 μm s$^{-1}$ (forward triangles, $N = 1202$), 3.2 μm s$^{-1}$ (circles, $N = 1039$), 6.4 μm s$^{-1}$ (pentagons, $N = 1129$), BE fit (dashed line, $\Delta x = 0.083$ nm, $k_{off}^0 = 2.8E-17$ s$^{-1}$). Weak state (red) in 10 mM EDTA (markers as before): 0.2 μm s$^{-1}$ ($N = 664$), 0.4 μm s$^{-1}$ ($N = 767$), 0.8 μm s$^{-1}$ ($N = 953$), 1.6 μm s$^{-1}$ ($N = 922$), 3.2 μm s$^{-1}$ ($N = 861$), 6.4 μm s$^{-1}$ ($N = 1007$), BE fit (dashed-dotted line, $\Delta x = 0.071$ nm, $k_{off}^0 = 0.011$ s$^{-1}$). **d** Ca$^{2+}$ titration experiment with SdrG B1 immobilized on a surface in varying Ca$^{2+}$ concentrations, starting from EDTA 10 mM to Ca$^{2+}$-free buffer in which all B1 unfolding events show the weak state. At 10 pM Ca$^{2+}$ the strong state starts to appear constituting the majority of unfolding events at 1000 pM Ca$^{2+}$. There are almost no unfolding events in an intermediate regime ($N = 995$). **e** Affinity estimate of SdrG B1 from combined Ca$^{2+}$ titration experiments, showing the fraction of all curves with B domain unfolding events in the strong state. A four-parameter logistic regression fit (red line) yields an inflection point of 0.144 nM, pointing to a sub-nM K$_D$ of SdrG B1 for Ca$^{2+}$ in the weak state, albeit concentration uncertainties (error bars as trace Ca$^{2+}$ uncertainty in buffer and 1% dilution error, $N = 1703$) in the sub-nM range are very high

**Incapacitating Ca$^{2+}$-binding sites lowers SdrG B1 stability.** Clearly, the addition of EDTA, i.e., the removal of Ca$^{2+}$, weakened the B1 domain. However, we could not discern if all, or only a fraction of the three Ca$^{2+}$ ions were removed. To determine how and which Ca$^{2+}$ ions were crucial to the stability, mutants lacking the amino acids required to coordinate each Ca$^{2+}$ were produced (hereafter Ca1KO, Ca2KO, Ca3KO, respectively), shown in Figs. 3a, b. To map the interplay between the loops, additionally all permutations of mutants with two Ca$^{2+}$ sites deleted, leaving only a single Ca$^{2+}$ bound, were created (Ca1,2KO; Ca2,3KO; Ca1,3KO, overview in Supplementary Fig. 1b). All mutants were probed in a single AFM-SMFS experiment using the same cantilever in both 10 mM EDTA and 10 mM Ca$^{2+}$, to compare absolute stabilities[25,32].

Results are shown in Fig. 3c (detailed distributions in Supplementary Fig. 6): Ca1 is coordinated by the largest number of negatively charged amino acids side chains (Fig. 3a), which intuitively would make it the most important, and a likely candidate to stay bound in chelating conditions. Interestingly, the Ca1KO mutant was only half as strong compared with the WT in Ca$^{2+}$. Ca2KO was only ~ 10% weaker than the WT in Ca$^{2+}$. For both Ca1KO, Ca2KO the weak state in EDTA remained at WT strength. Ca3KO showed the most drastic change both in Ca$^{2+}$ and EDTA, as unfolding forces dropped by an order of magnitude. When incapacitating two Ca$^{2+}$-binding sites at a time (Fig. 3d) Ca1,2KO behaved similar to the Ca1KO mutant, hinting that Ca2 could still be occupied by a Ca$^{2+}$ ion as it interacts with parts of the peptide backbone and an aspartic acid

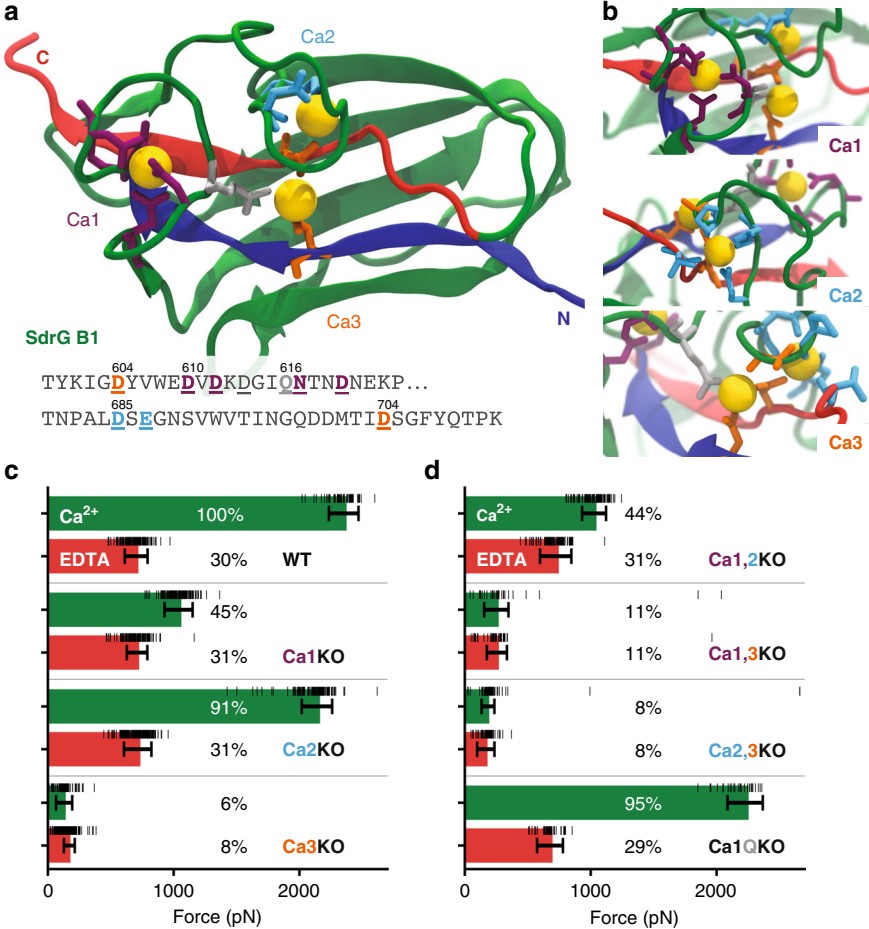

**Fig. 3** $Ca^{2+}$ loop three is most important to SdrG B1 domain stability. **a** Equilibrated homology model of SdrG B1, with N- and C-terminal ß-strands marked in blue and red, respectively. $Ca^{2+}$ coordinating amino-acid side chains shown as sticks. $Ca^{2+}$ binding sites one to three are marked (Ca1 purple, Ca2 cyan, Ca3 orange). A conserved glutamine bridge (light gray) connects the Ca1 loop to Ca3. Amino acids mutated to remove respective loops are underlined in the sequence shown. **b** Closeup of each $Ca^{2+}$ binding loop, including non-sidechain coordinating residues in stick representation. **c**, **d** Comparison of absolute unfolding forces of mutated SdrG B1 in 10 mM EDTA (red) and 10 mM $Ca^{2+}$ (green) with a single cantilever, also given in percentage of the WT strong state. Errors are the full-width at half maximum of the BE fits for each unfolding force distribution (see Supplementary Fig. 6), underlying raw force datapoints are shown as black horizontal lines. The single loop knockouts in (**c**) show that Ca3 is crucial for overall stability and most likely remains bound in the weak state. Once Ca3 is removed, weak state forces drop to only 6–8% of WT strength. Removing Ca1 or Ca2 or both (**d**) supports this as the weak state remains at 31% of WT. The glutamine bridge (Ca1QKO) seems to be of minor importance for the overall domain stability

of Ca3. These contacts might be sufficient for coordination (see detailed coordination sites in Supplementary Fig. 7). Ca1,3KO and Ca2,3KO were drastically weaker than the WT, comparable to Ca3KO. The removal of a conserved glutamine bridge (Ca1QKO) between Ca1 and Ca3 only led to a minor decrease in domain strength.

In summary, Ca3 is most crucial for overall B domain stability, essential to establishing the 600 pN weak state, and most likely stays bound in the presence of EDTA. Adding Ca2 increases the stability to over 1 nN, whereas adding Ca1 boosts it to over 2 nN. Even in very dilute $Ca^{2+}$, well below the $K_D$ in the titration series, we never observed more than a handful of events in a force range comparable to Ca1KO and Ca2KO (see Fig. 2d). Instead, SdrG B1 immediately occupied the strongest state. Thus, we propose that binding of Ca1 and Ca2 must be highly cooperative.

**Homologous B domains show similar unfolding forces.** SdrG contains a second B domain (B2), whose sequence is 45% identical to B1 (alignment, see Supplementary Fig. 1). An equilibrated homology model is shown in Fig. 4a. The crystalized SdrD B1 domain, shown equilibrated in Fig. 4b, was investigated, too.

When measured in 10 mM $Ca^{2+}$, SdrG B2 and SdrD B1 showed a similar, 2 nN stability. However, in EDTA a weak unfolding event with their expected contour length appeared only rarely. Most curves contained no discernible unfolding peak (above our detection limit around 20 pN), hinting at a complete unfolding of the domains, a marked difference from to the mere weakening of SdrG B1. The $Ca^{2+}$-EDTA switching, for SdrD B1 (Fig. 4c) thus resulted in very few weak events detected in EDTA, which showed a bimodal unfolding force distribution that was described well by a superposition of two BE fits (Eq. 1, Fig. 4c).

Given their highly similar structure, this result was unexpected. Subtle differences in B domains must give them diverging properties. When comparing all three domains, further differences emerged: the $Ca^{2+}$ affinity of SdrD B1 was slightly lower than for SdrG B1 and lowest for SdrG B2 (fraction of all curves with folded domain, i.e., strong and weak in low $Ca^{2+}$: SdrG B1 ~ 87%, SdrD B1 ~ 80%, SdrG B2 ~ 33%, Fig. 4d). Intriguingly, SdrG B1 and adjacent B2 have clearly separated regimes at which they switch to their strongest state. A comparison of the absolute mechanostabilities of SdrG B1, SdrG B2, and SdrD B1 was conducted with a single cantilever[25], using ClfB as a handle. The results are depicted in Fig. 4e. SdrD B1 and SdrG B2 exhibit

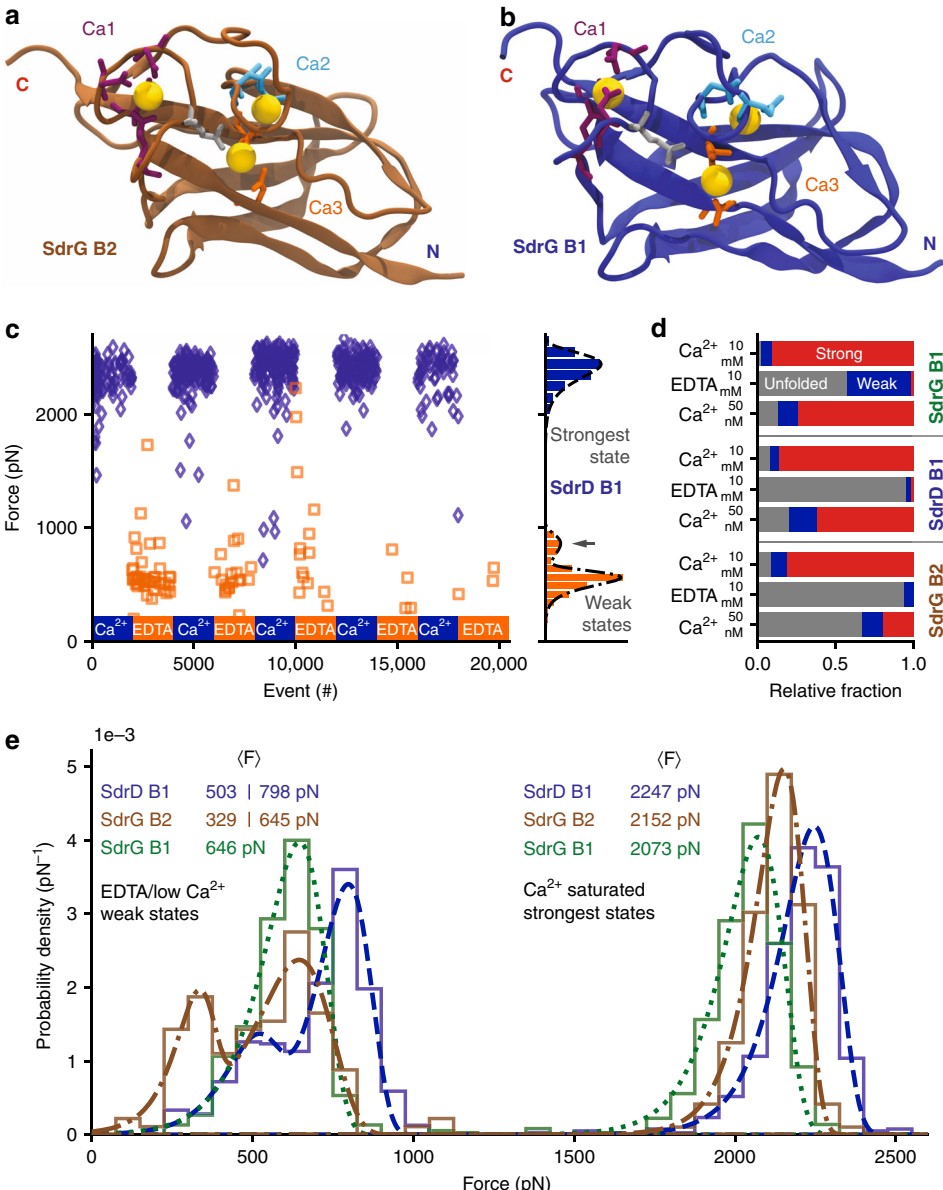

**Fig. 4** SdrG B2 and SdrD B1 are even stronger than SdrG B1 and have different $Ca^{2+}$ affinities. **a** Equilibrated homology model of SdrG B2 (brown) with $Ca^{2+}$ (yellow) binding loops one to three marked, coordinating amino-acid side chains shown as sticks (coloring as in Fig. 3a). **b** Equilibrated crystal structure of SdrD B1 (blue, PDB 4JDZ). **c** Cycling of SdrD B1 between 10 mM $Ca^{2+}$ and 10 mM EDTA, a strong state in $Ca^{2+}$ over 2000 pN emerges. The weak state is bimodal peaking at ~ 550 pN and ~ 850 pN (gray arrow), fit with a superposition of two BE p(F) functions (see methods). Very few unfolding events appear in 10 mM EDTA, thus the $Ca^{2+}$-chelation must be sufficient to completely unfold most SdrD B1 domains. The strong state is also bimodal in low $Ca^{2+}$ conditions see Supplementary Fig. 8. **d** Relative fraction of states in different buffers: 10 mM $Ca^{2+}$, 10 mM EDTA, and 50 nM $Ca^{2+}$ (applied after 10 mM EDTA) for SdrG B1, B2 and SdrD B1 compared with a single cantilever: no unfolding event (unfolded protein, gray), weak state (< 1000 pN, blue), strong state (>1500 pN, red). Almost all SdrD B1 and SdrG B2 domains are unfolded in EDTA and only a fraction refolds in low $Ca^{2+}$. SdrG B2 has the lowest affinity for $Ca^{2+}$ as it shows the least folded events in low $Ca^{2+}$. **e** Comparison of relative unfolding forces of all B domains at 1.6 µm s$^{-1}$ with a single cantilever. The strongest state of SdrD B1 (blue, dashed line) is the most mechanostable, followed by SdrG B2 (brown, dash-dotted line), then SdrG B1 (green, dotted line). The weak states of SdrD B1 and SdrG B2 are bimodal and best described with a superposition of two BE fits, whereas SdrG B1 only has a single weak state

bimodal unfolding force distributions in the weak state. The strong states of these domains are also bimodal in low (~ 50 nM) $Ca^{2+}$ concentrations (not shown here, see Supplementary Fig. 8), hinting at a separable, stepwise binding of $Ca^{2+}$. In saturating 10 mM $Ca^{2+}$ conditions, all domains have a unimodal unfolding force distribution, with SdrD B1 being the strongest, followed by 4% weaker SdrG B2, and 8% weaker SdrG B1. Although SdrG B2 has a lower affinity for $Ca^{2+}$ than SdrG B1 it is clearly stronger,

demonstrating that B domain $Ca^{2+}$ affinity is not correlated with its mechanostability.

## Discussion

Why have B domains evolved these exceptional mechanical stabilities? Their extreme mechanostability may be rationalized in context of the ligand-binding region, which they connect to the bacterium. In the case of SdrG, the mechanical stability of the

interaction between the N2 and N3 domains binding their Fgß target is independent of the B domains, as they can be deleted from the construct without lowering the interaction rupture force[9]. Contrary to DLL adhesins, recently studied thioester domain (TED) adhesins attach to their human targets through a covalent isopeptide bond[33–35]. Covalent bonds are mechanically stronger and irreversible compared with the non-covalent, spontaneously reversible, DLL attachment ($K_D$ for SdrG N2N3: Fgß ~ 400 nM)[5]. Isopeptide bonds are covalent amide bonds between amino-acid side chains that stabilize a fold, or connect two proteins. TEDs and collagen-binding MSCRAMMs are linked by Ig-like and $Ca^{2+}$ coordinating folds such the Spy0128 pilus of *S. pyogenes*, which contain intramolecular isopeptide bonds[36,37]. These block the mechanical extension of Spy0128[38]. Notably, another Ig-like domain from *S. pyogenens* fibronectin binding protein (fba2) follows a remarkably similar fold to the B domains here (structural alignment in Supplementary Fig. 9). Their most striking difference is that fba2 is stabilized by the isopeptide bond locking the N and C-terminal ß-sheet together. The SpyTag/Catcher covalent labeling system was derived from this system[39]. In SdrG B1 and its homologs, the coordinated $Ca^{2+}$ ions are covering these strands and give the domain extraordinary mechanostability. The rigidity of B regions, especially those containing isopeptide bonds, such as the collagen adhesin Cna of *S. aureus*[40], has been proposed to project the ligand-binding region away from the bacterial surface toward the host[37,41,42], which could also be a function of the B domains investigated here. Isopeptides, as covalent bonds, are stronger than the B domain fold, making them completely resistant to unfolding, ultimately resulting in a stiff, rigid stalk. In contrast, B domains can un- and refold. Both systems achieve high mechanical stability, however, they differ in what is best addressed as malleability. SdrG B domains can act as a "mechanical shock dissipater" under tension, as previously proposed for pili domains by Echelman et al.[23]. Domains unfold to buffer transient stress, e.g., caused by shear flow, on the ligand-binding region at the tip and regenerate when tension recedes.

At physiological $Ca^{2+}$ concentrations (free $Ca^{2+}$ ions in human blood on the order of 1 mM[43]) at least SdrG B1 would be found almost exclusively in its strong state. However, the mechanical stabilities of the strongest state of SdrG B1 and B2 are on the order of the interaction of the SdrG N2N3 domains with fibrinogen. Sometimes the interaction is not sufficient to unfold the B domain at force loading rates around $10^4$ pN s$^{-1}$ – in clear conflict with the proposed shock dissipater by unfolding hypothesis. The BE fits (see Supplementary Fig. 10) show that the SdrG B1 domain's unfolding force dependency on the force loading rate has a less steep slope than the SdrG adhesin. Extrapolating from this range, the B domains would reliably be weaker than the N2N3 receptor–ligand interaction at higher force loading rates exceeding $10^6$ pN s$^{-1}$. In this range, at least SdrG B1 could fulfill a shock dissipater function. One could speculate that SdrG B1 only unfolds when rapid changes in load are stressing the SdrG adhesin, while letting slow changes in force act on the tip adhesin, thus acting as a low-pass filter for stress: a strong-and-sudden load dissipater.

Many MSCRAMM adhesins contain more than one B domain (e.g., SdrD has five B domains in total). Previous work suggested that these have different individual $Ca^{2+}$ affinities[21,31]. In the case of SdrG B1 and B2 have comparable unfolding forces, yet different $Ca^{2+}$ affinities, despite 45% sequence identity and high similarity in the $Ca^{2+}$ binding sites. Different B domains in the same adhesin thus may have specifically tuned functions. The varied $Ca^{2+}$ affinities may control their mechanical strength, ensuring that one domain is preferentially in the weak, while another one occupies the strong or an intermediate state, as in the case of SdrD B1 and SdrG B2, whose intermediate state strengths would be ideal to dissipate stress.

The mechanism that governs these extreme mechanical unfolding forces is clearly dependent on the presence of $Ca^{2+}$. The coordination of Ca3, connecting the parallel very N and C-terminal, closing ß-strands (see Fig. 3a in blue and red), is most crucial to overall B domain mechanostability. The molecular mechanism governing the comparable unbinding forces of the tip adhesin of SdrG N2N3:Fgß relies on the confined alignment of the backbone hydrogens bonds (H bonds) between the target peptide and the enclosing locking strand in a shear geometry[9]. Analogously, one could propose that such an H bond-based mechanism stabilizes B domains. Indeed, the equilibration simulations show only few H bond contacts between the N and C-terminal ß strands, with most of them at the very C-terminus, below the Ca1 loop site. This geometry may change upon force application, but the mechanism that gives B domains their exceptional mechanostability most likely differs from SdrG:Fgß, in that the $Ca^{2+}$ electrostatically protects the H bonds from breaking and locks them in a shear geometry. Alternatively, the coordination of $Ca^{2+}$ ions may serve as a network though which forces propagate, diverting the load from the closing N- and C-terminal ß-sheets.

B domains are the mechanically strongest proteins examined to date, surpassing the stability of previously probed folds by at least a factor of two. B domains draw their stability from the coordination of $Ca^{2+}$ ions, which are in some cases required for their refolding process. B domain $Ca^{2+}$-dependent force resilience offers a blueprint to design extremely stable biomaterials with adjustable force response. In particular, SdrD B1 and SdrG B2 domain folding is tunable: from completely unfolded in EDTA, through a weak state in low $Ca^{2+}$ unfolding around 500–800 pN, to over 2 nN strong in high $Ca^{2+}$. Each state can be induced through a $Ca^{2+}$/EDTA stimulus, respectively. Such properties may be fundamentally interesting as protein folding models that do not require aggressive denaturants to unfold, and more practically useful, e.g., in a stimuli-responsive protein hydro-gel[44,45]. A network of B domains could withstand extreme forces when contracted and folded in $Ca^{2+}$ but change into a flexible, extended polypeptide mesh when exposed to a $Ca^{2+}$-chelating agent. Furthermore, SdrG B2 and SdrD B1 may be used for $Ca^{2+}$ sensing, as their folding upon $Ca^{2+}$ binding could be read out by monitoring FRET of dyes attached at their N- and C-termini.

Finally, the role of B domains in pathogen adhesion remains debatable. Roles such as extendable springs that stretch and contract, or a shock dissipater have been suggested[41,46]. However, the high SdrG B domain unfolding forces overlap with their respective receptor ligand unbinding at physiological $Ca^{2+}$ concentrations and force loading rates around $10^4$ pN s$^{-1}$, which prevent them from being reliable load dissipaters in this range. The weak state is ideally suited for this task, and at higher force loading rates, so are the strong states. It remains to be examined how B domains interact with each other[46], respond to constant forces or low force loading rates, as well as changes in pH, temperature or ionic strength, and what force the B domains exert when folding in the presence of $Ca^{2+}$. The calcium-dependent, ultrahigh mechanical stability of the B domain fold demonstrates to which extreme physical regimes pathogens evolved to invade their hosts.

## Methods

**Chemicals**. All chemicals used were supplied by Carl Roth (Karlsruhe, Germany) or Sigma-Aldrich (St. Louis, MO, USA) if not specified explicitly.

**Gene construction**. The *Dictyostelium discoideum* 4th filamin fingerprint domain (ddFLN4, UniProt: P13466, residues 549–649, with the internal cysteine mutated to

serine), the *Staphylococcus epidermidis* SdrG N2N3, B1, and B2 domain genes (UniProt: Q9KI13), the *Staphylococcus aureus* ClfB N2 and N3 domains (UniProt: Q7A382); the SdrD B1 domain (from PDB 4JDZ with incomplete sequence, complete sequence in GenBank: WP_000934487 or obsolete UniProt entry: E5QTK7) were synthesized codon-optimized for expression in *Escherichia Coli* as linear DNA fragments (GeneArt – ThermoFisher Scientific, Regensburg, Germany) including suitable overhangs for Gibson assembly. Genes were inserted into pET28a Vectors with a hexahistidine-, ybbr-tag and in some cases a sortase motif via Gibson assembly [47] (New England Biolabs, MA, USA). All point mutations, deletions, or additions of amino acids in all systems were created through poly-merase chain reactions (Phusion Polymerase, New England Biolabs, MA, USA) with appropriate primers and finally blunt end ligation cloning using T4 Ligase (Thermo Scientific, MA, USA). Resulting open reading frames of all constructs were verified by DNA sequencing (Eurofins Genomics, Ebersberg, Germany). All protein sequences of constructs used in this study are listed in the Supplementary Information.

Important plasmids were deposited with Addgene (www.addgene.org) and are available through the following AddgeneIDs:

**Protein expression and purification**. All proteins were expressed in *E. Coli* NiCo21(DE3) (New England Biolabs, MA, USA). Bacterial starter cultures of 5 mL Lysogeny broth (LB) medium containing 50 µg mL$^{-1}$ Kanamycin, were inoculated and grown overnight at 37 °C. These were added into in 100–200 mL of ZYM-5052 autoinduction media[48] containing 100 µg mL$^{-1}$ Kanamycin and grown for 6 h at 37 °C and cooled down, continuing overnight at 18 °C. For small-scale protein production, 8 mL cultures in ZYM-5052 autoinduction media were grown at 37 °C overnight. Bacteria were harvested by centrifugation at 8000 *g*, the supernatant was discarded, pellets were stored at −80 °C until purification.

All purification steps were performed at 4–8 °C. The bacterial pellet was resuspended in lysis buffer (50 mM TRIS, 50 mM NaCl, 5 mM MgCl$_2$, 0.1% (v/v) Tween-20, 10% (v/v) glycerol, pH 8.0) with 100 µg mL$^{-1}$ lysozyme (Carl Roth, Karlsruhe, Germany). Cells were lysed by sonication (Sonoplus GM 70, with a microtip MS 73, Bandelin, Berlin, Germany). Insoluble parts were separated by centrifugation at > 40,000 *g* for at least 30 min. The supernatant was sterile filtered (0.45 µm, then 0.22 µm pore size), adjusted to contain 20 mM imidazole, and then loaded onto a Ni-NTA column (HisTrap FF 5 mL on a Äkta Start system, both GE Healthcare, MA, USA) for HIS-Tag purification and washed extensively (25 mM TRIS, 500 mM NaCl, 20 mM imidazole, 0.25% (v/v) Tween-20, 10% (v/v) glycerol, pH 7.4). The protein was eluted in the same buffer, only different in containing 200 mM imidazole and being at pH 7.8. Protein containing fractions were concentrated in centrifugal filters (Amicon, Merck, Darmstadt, Germany), exchanged into measurement buffer (TBS: 25 mM Tris, 75 mM NaCl, pH 7.4) by desalting columns (Zeba, Thermo Scientific, MA, USA), adjusted to 10% (v/v) glycerol, and frozen in aliquots in liquid nitrogen to be stored at −80 °C until thawed for experiments. Protein concentrations were determined by spectrophotometry at 280 nm with typical final concentrations of 30–1000 µM (NanoDrop 1000, Thermo Scientific, MA, USA).

**AFM sample preparation**. More detailed AFM-SMFS protocol have been pub-lished previously[16,49]. In brief, AFM Cantilevers (Biolever Mini AC40TS, Olympus, Tokyo, Japan) and 24 mM diameter cover glass surfaces (Menzel Gläser, Braunschweig, Germany) were modified with Aminosilane.

Glass surfaces: Glass surfaces were cleaned by sonication in 50% (v/v) 2-propanol in ultrapure H$_2$0 for 10 min, rinsed with ultrapure H$_2$0, and further cleaned and oxidized in 50% (v/v) H$_2$0$_2$ and 50% (v/v) of 30% (v/v) sulfuric acid for 20 min. Surfaces were washed in ultrapure H$_2$0, then ethanol. Surfaces were silanized by soaking in a solution of: 2% (v/v) (3-aminopropyl) dimethylethoxysilane (ABCR, Karlsruhe, Germany), 88% (v/v) ethanol, and 10% (v/v) ultrapure H$_2$0 under gentle shaking for 1 h. Again, followed by two rinsing steps in ethanol, then rinsed in ultrapure H$_2$0, and afterwards dried in a gentle stream of nitrogen. Finally, surfaces were baked at 80 °C for 45 min. Glass surfaces were stored under Argon and typically used within 1 month.

Cantilevers: Following 15 min of UV-Ozone cleaning (UVOH 150 LAB, FHR Anlagenbau GmbH, Ottendorf-Okrilla, Germany), cantilevers were silanized, submerged in 1 mL (3-aminopropyl)-dimethylethoxysilane (APDMES, abcr, Karlsruhe, Germany) mixed with 1 mL ethanol and 5 µL ultrapure H$_2$0 for 5 min. Each cantilever was rinsed in ethanol and subsequently in ultrapure H$_2$0. Finally, cantilevers were baked at 80 °C for 1 h to be stored overnight under Argon and used in the following steps the next day.

Both glass surfaces and cantilevers were covered with 5 kDa heterobifunctional α-Maleinimidohexanoic-PEG-NHS (Rapp Polymere, Tübingen, Germany) dissolved in 50 mM HEPES (pH 7.5) at 25 mM (125 mg mL$^{-1}$) for 30 min. After rinsing surfaces and cantilevers in ultrapure water, 1 mM coenzyme A (in 50 mM sodium phosphate pH 7.2, 50 mM NaCl, 10 mM EDTA buffer) was applied to both for at least 1 h. CoA functionalized surfaces and cantilevers stored in coupling buffer (50 mM sodium phosphate pH 7.2, 50 mM NaCl, 10 mM EDTA buffer) at 4 °C were stable for >4 weeks.

When different protein constructs were compared with a single cantilever, up to 10 spatially separated spots were created using a silicone mask (CultureWell reusable gaskets, Grace Bio-Labs, Bend, OR, USA), cleaned by sonication in

isopropanol and ultrapure H$_2$0, dried in a gentle stream of nitrogen, heated to 60 °C and securely pressed onto a silanized microscopy slide (76 × 26 mM, Carl Roth, Karlsruhe Germany). Pegylation and CoA coupling in individual wells was achieved following identically to the protocol described above[25].

These steps yielded cantilevers and surfaces covalently coated in PEG-CoA. Cantilevers and surfaces were rinsed in ultrapure water. Protein functionalization was achieved by covalently pulling down proteins via their ybbr-tag to CoA by the SFP enzyme coupling. The proteins of interest were diluted into TBS150 (25 mM Tris, 150 mM NaCl, pH 7.4) supplemented with 10 mM MgCl$_2$. Cantilevers were typically incubated with 50 µM of protein of interest and 3 µM Sfp phosphopantetheinyl transferase (SFP) for at least 1 h. The glass surfaces were incubated with 5–15 µM of protein construct of interest and 1–2 µM SFP for 30–60 min, depending on the desired surface density. Both samples were rinsed extensively with at least 60 mL measurement buffer (TBS75: 25 mM Tris, 75 mM NaCl, pH 7.4) buffer before experiments.

**AFM-SMFS**. AFM-SMFS data were acquired on a custom-built AFM operated in closed loop by a MFP3D controller (Asylum Research, Santa Barbara, CA, USA) programmed in Igor Pro 6 (Wavemetrics, OR, USA). Experiments were conducted at room temperature (approximately 25 °C). Cantilevers were briefly (<200 ms) and gently (<200 pN) brought in contact with the functionalized surface and then retracted at constant velocities ranging from 0.4, 0.8, 1.6, 3.2 to 6.4 µm s$^{-1}$ for a dynamic force spectrum, otherwise and for titration experiments a standard velocity of 1.6 µm s$^{-1}$ was used. After each curve acquired, the glass surface was moved horizontally by at least 100 nm to expose an unused, fresh surface spot. Typically, 50,000 – 100,000 curves were recorded per experiment. When quanti-tative comparisons of absolute forces were required, a single cantilever was used to move between multiple spatially separated spots to be probed on the same surface (created using the protocol described above). To calibrate cantilevers, the inverse optical cantilever sensitivity (InvOLS) was determined as the linear slope of the most probable value of typically 40 hard (>2000 pN) indentation curves. Canti-levers spring constants were calculated using the equipartition theorem method with typical spring constants between 70–160 pN nm$^{-1}$[50,51]. A full list of calibrated spring constants from experiments presented in this work is provided in the supplementary methods, as the stiffness of the pulling handle, i.e., the cantilever, may influence the complex rupture and domain unfolding forces measured.

**Calcium titration experiments**. Buffer made from ultrapure water (resistivity 18.2 MΩ cm, arium pro, Sartorius, Goettingen, Germany), TRIS and NaCl (both Carl Roth, Karlsruhe, Germany) contained too much Ca$^{2+}$ to reliably measure Ca$^{2+}$ binding (Ca$^{2+}$-free buffer from ultrapure water already showed over 50% of SdrG B1 domains in strong, Ca$^{2+}$-bound state, even though the sample had been Ca$^{2+}$ depleted with at least 10 mM EDTA before). Instead, water (Ultra Quality HN68.2, Carl Roth, Karlsruhe, Germany) containing ≤10 ppt Ca$^{2+}$ (≤10 parts per trillion, i.e., ≤10E−12 kg kg$^{-1}$, or ≤10E−9 g L$^{-1}$, which computes to ≤0.250 nM for a Ca$^{2+}$ ion) according to the manufacturer was used. Buffering was achieved by dissolving a Ca$^{2+}$-free PBS tablet (Thermo Fisher Scientific, MA, USA). B domains immobilized on the surface were Ca$^{2+}$ depleted with at least 10 mM EDTA and after repeated rinsing in Ultra Quality Ca$^{2+}$-free PBS, the titration was started. For each concentration, thousands of curves were acquired, before the surface was rinsed with the next Ca$^{2+}$ concentration. As the discerning between strong and week state should not depend on cantilever stiffness, multiple experiment data using identical buffers were pooled to build statistics.

**SMFS data analysis**. Data analysis was carried out in Python 2.7 (Python Software Foundation)[52–54]. Laser spot drift on the cantilever relative to the calibration curve was corrected via the baseline noise (determined as the last 5% of datapoints in each curve) for all curves and smoothed with a moving median (windowsize 300 curves). The InvOLS for each curve was corrected relative to the InvOLS value of the calibration curve.

Raw data were transformed from photodiode and piezo voltages into physical units with the cantilever calibration values: The piezo sensitivity, the InvOLS (scaled with the drift correction) and the cantilever spring constant (k).

The last rupture peak of every curve was coarsely detected and the subsequent 15 nm of the baseline force signal were averaged and used to determine the curve baseline, which was then set to zero force. The origin of molecule extension was then set as the first and closest point to zero force. A correction for cantilever bending, to convert extension data in the position of the cantilever tip was applied. Bending was determined through the forces measured and was used on all extension datapoints (x) by correcting with their corresponding force datapoint (F) as $x_{corr} = x - F/k$.

To detect unfolding or unbinding peaks, data were denoised with total variation denoising (raw, not denoised, data shown in plots)[55,56], and rupture events detected as significant drops in force relative to the baseline noise. A three-regime polymer elasticity model by Livadaru et al.[57] was used to model the behavior of contour lengths freed by unfolding events and transformed into contour length space [58] (Livadaru model parameters were: stiff element $b = 0.11$ nm and bond angle $γ = 41°$). A quantum mechanical correction was used to account for peptide bond stretching at high forces [59]. Especially at forces larger than 1 nN, this correction was essential to be able to fit the data to polymer elasticity models

accurately. Peaks were assigned their contour length in diagrams assembled through Kernel Density Estimates (KDE) of the contour length transformed force-extension data. The KDE bandwidth was chosen as 1 nm. The loading rate was fitted as the linear slope of force vs. time of the last 4 nm preceding a peak.

For single BE model at a given force loading rate r (determined as most probable loading rate from all unfolding events through a KDE) with the parameters $\Delta x$ and $k_{off,0}$, the probability density $p(F, r, \Delta x, k_{off,0})$ to unfold at a given force F was fit to a normalized force histogram. For a superposition of two BE fits as in Fig. 4, the unfolding force histogram was fit with Eq. 1:

$$p_{total}(F, q, r_1, \Delta x_1, k_{off}{}^0{}_1, r_2, \Delta x_2, k_{off}{}^0{}_2) = q \times p_{BE1}(F, r_1, \Delta x_1, k_{off}{}^0{}_1)$$
$$+ (1-q) \times p_{BE2}(F, r_2, \Delta x_2, k_{off}{}^0{}_2)$$

$$(1)$$

Force loading rates $r_1$ and $r_2$ were assigned at a force $f_{critical}$ at the minimum value of $p_{total}$ between the maxima of both BE fits and then assigned to BE1 and BE2, as force loading rate and unfolding force correlate in a constant velocity experiment. The relative weight of each distribution was q for BE1 and $(1-q)$ for BE2 with $0 < q < 1$.

For dynamic force, spectra rupture force histograms for the respective peaks and dynamic force spectra were assembled from all curves showing B domain unfolding, or (if applicable) a specific fingerprint domain, and/or a clean complex rupture event. The most probable loading rate of all complex rupture or domain unfolding events was determined with a KDE, bandwidth chosen through the Silverman estimator[60]. This value was used to fit the unfolding or rupture force histograms with the BE model for each pulling velocity[61,62]. Errors in all diagrams are given as the asymmetric full-width at half maximum (FWHM) of each probability distribution. A final fit was performed through the most probable rupture forces and loading rates for each pulling velocity to determine the distance to the transition state $\Delta x_0$ and natural off-rate at zero force $k_{off}{}^0$.

**Homology models and simulations.** Homology models for the SdrG B1, SdrG B2 domain, and SdrG N2N3-B1-B2 construct were created using Modeller 9.19[63,64]. Template file for the B domains was PDB 4JDZ (SdrD B1 domain) and for the SdrG N2N3-B1-B2 construct PDB 1R17 (SdrG N2N3) was added.

Model structures were equilibrated in water using the NAMD[28] molecular dynamics package with setups created by VMD[65] plugin QwikMD[27]. The CHARMM36[66] force field and TIP3[67] water model were used in all simulations. Structures were centered in a water box 15 Angstrom larger than the protein's longest dimension, NaCl was added to 150 mM. Minimization (2000 steps), then Annealing (0.29 ns, temperature rise 60 K to 300 K, 1 atm pressure, protein backbone restrained), then equilibration (1 ns, temperature 300 K, 1 atm pressure, protein backbone restrained), then MD simulation (temperature 300 K, 1 atm pressure, no restraints) were performed in the NpT ensemble. Final simulations were run for at least 100 ns for individual B domains and at least 30 ns for SdrG N2N3-B1-B2.

Simulation parameters were: a distance cut-off of 12.0 Å was applied to short-range, non-bonded interactions, and 10.0 Å for the smothering functions. Long-range electrostatic interactions were treated using the particle-mesh Ewald[68] method. The pressure was maintained at 1 atm using Nosé-Hoover Langevin piston[69,70]. The equations of motion were integrated using the reversible reference system propagator algorithm (r-RESPA) multiple time step scheme to update the short-range interactions every 1 steps and long-range electrostatics interactions every 2 steps. The time step of integration was chosen to be 2 fs for all simulations. The temperature was maintained at 300 K using Langevin dynamics.

## Data availability

Data supporting the findings of this manuscript are available from the corresponding authors upon reasonable request. A reporting summary for this article is available as a Supplementary Information file. The source data underlying Figs. 1e, f, 2b-e, 3c, d, and 4c-e are provided as a Source Data file.

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

## Acknowledgements
We gratefully acknowledge funding from the Deutsche Forschungsgemeinschaft DFG (SFB1032, GA309/11). We thank Ellis Durner, Markus A. Jobst, Wolfgang Ott, and Tobias Verdorfer for work on instrumentation and/or surface chemistry; Rafael C. Bernardi for pointers concerning modeling.

## Author contributions
H.E.G. and L.F.M. designed the research. T.N. cloned and expressed constructs, and prepared surface chemistry reagents. E.M.U. and L.F.M performed and analyzed experiments. H.E.G. and L.F.M. wrote the manuscript.

## Additional information

**Competing interests:** The authors declare no competing interests.

