## [Peer Review File · Nature Communications]

Reviewers' Comments:

Reviewer #1:

Remarks to the Author:

Understanding the adhesion and mechanics of staphylococcal surface proteins is a hot topic with multiple medical and technical implications. In this interesting manuscript the authors unravel the mechanical stability of Ig-like folds (B domains) of staphylococcal adhesins, a longstanding challenge. The results are solid as well as the conclusions, no doubt the work will interest a broad audience.

1) last sentence of abstract: hinting at its role in for pathogen adhesion; I'm unclear how these key findings are that important for pathogenicity; the authors should qualify (remove the sentence) or elaborate further in conclusions.

2) Intro, line 24. Worth to discuss Herman, mBio: In vivo, the B region of Cna is required for strong ligand binding and has been found to function as a spring capable of sustaining high forces of 1,2 nN. This previously undescribed mechanical response of the B region is of biological significance as it provides a means to project the A region away from the bacterial surface and to maintain bacterial adhesion under conditions of high forces.

3) Double peaks in fig 2a are reminiscent of the double peaks observed on live bacteria with SdrG (Herman, Mol Mic), worth to comment on this.

4) Ca plays a key role, but were other divalent tested?

5) Conclusions. Comment on the link with the Cna paper above. Also a couple of staph adhesins show catch bond like behavior, e.g. the bond between ClfA and immobilized Fg (Herman Pnas). To what extent the role of Ca could be loading rate dependent and be linked to catch bond mechanisms?

Reviewer #2:

Remarks to the Author:

Milles et al. present a very interesting paper in which they characterize how B domains stabilize by the coordination of Ca²⁺ ions. It is observed that the SdrD B1 and SdrG B2 domain folding is modulated from completely unfolding in EDTA chelating the Ca²⁺ ions to an extremely stable fold in higher Ca²⁺ concentrations. The possibility to modulate protein folding and stability by 'Ca²⁺ stimulation' is biotechnologically extremely interesting and promising. As such the authors outline that more complex structures involving B domains withstand extreme forces when contracted and folded in Ca²⁺, and upon exposure to Ca²⁺-chelating agents change into a flexible, extended polypeptide mesh. However, the exceptional mechanical and kinetic stability and calcium-tunable folding and force response suggest B domains as very promising protein design targets. I found the paper to be extremely well written, thought through and presented. It was a pleasure to read. The experiments are done at the highest standards (e.g., by leaders of the field) and the biological and biotechnological relevance of the paper is very high. I anticipate it will inspire various scientific fields and disciplines.

I have a few comments I would like to address to the authors

The authors write that the B2 domains may be used to sense Calcium. However. Calcium sensors of chemical and biological origin are well known and frequently used in for example live cell fluorescence imaging of calcium activities (gcamp etc). Possibly the authors could further line out why it may be needed to use a mechanical readout to sense calcium.

The authors write that 'It remains to be examined how B domains interact with each other, respond to constant forces or low force loading rates and what force the B domains exert when folding in the presence of Ca²⁺.' Certainly, studying the domain stability depending on the listed parameters is interesting. However, to better understand the pathogenic function of the domains contributing to bacterial adhesion I would also suggest to characterize the B domain stability in dependence of other physiological parameters including pH, temperature, or other salts. Particularly the pH can change extremely from about 2-10.

The authors write about the physiological Calcium concentration. Please provide range and reference for your system.

Please define the temperature at which the experiments were performed.

Please provide errors of the spring constants determined.

Reviewer #3:

Remarks to the Author:

The manuscript by Milles et al reports a fascinating atomic force spectroscopy study of the mechanical strength of beta domains of adhesins from pathogenic bacteria and its modulation by calcium. Prior to infection, bacteria such as *Staphylococcus aureus* adhere strongly to the host using highly specialized protein anchors that bind to receptors and need to withstand large pulling forces. In an earlier study, the same group, using AFM-based single molecule force spectroscopy (SMFS), determined that the molecular system mediating the adhesive interaction is unusually strong, with rupture forces exceeding 2000 pN (with a force of tens to a hundred pN being a typical strength of hydrogen bonds in biomolecular systems, for comparison). As immunoglobulin-like B domains are an integral part of the molecular system engaging in the adhesion (located upstream of the receptor-ligand part), the authors now asked the question how mechanically strong are these B domains. By attaching a single B domain at one end via a short linker to the tip of the AFM cantilever and the other end through N2N3 domains and the Fgbeta linker and some auxiliary reference modules to the glass substrate, they were able to directly pull on the B domain and measure its mechanical tensile strength. The SMFS results suggest that the B1 domain unfolds mechanically through two different pathways. The first pathways captured moderately large unfolding forces of around 600 pN and the second pathway reveals extremely high unfolding forces exceeding 2000 pN, suggesting a B1 domain with enormous mechanical stability. Moreover, the mechanical strength of this domain appears to be modulated by calcium ions in a concentration-dependent fashion, as calcium may occupy up to three binding sites within the domain. Mutations within these sites reveal that site # 3 is most important for the strength of the module. Similar SMFS measurements are then carried out for SdrG B2 domain from the same species and for the SdrD B1 domain from *Staph. aureus*. Essentially, all these B domains reveal extremely high mechanical stability with unfolding forces over 2 nN and also show more subtle differences in responding to calcium and in the proportion of "lower mechanical stability pathway". Overall the work is carried out with great attention to detail, particularly with respect to generating various constructs and mutants and carrying out a significant number of measurements at different loading rates.

However, I have three methodological observations and concerns that I wish the authors consider:

a) Including the ddFLN4 domain in the construct is clearly beneficial for identifying single molecule AFM recordings, as this domain provides an unambiguous mechanical unfolding fingerprint in force-extension curves. However, by itself, a construct which has the ddFLN4 domain only on one side of the B domain of interest (as used by the authors) cannot provide undisputable evidence that the B domain was subjected to stretching forces and unfolded completely in a given experiment. The contour length increment following the first large unfolding peak being consistent with the length of the B domain helps to suggest that this unfolding peak originated from the B

domain, but by itself, does not provide direct evidence. For this, one needs to have the ddFLN4 domain (or another fingerprint generating domain) also on the other side of the B domain in the protein construct.

b) As mentioned above, contour length increments following the first unfolding peak attributed to the B domain are very important but the data about distributions (pdf) of this length increment is not included in the manuscript. I am sure many readers would like to see those distributions (for all B domains and under various calcium conditions) to be shown directly as accompanying unfolding force distributions as they are equally important to this study.

c) It would be useful to show, either in the main manuscript or in supporting information, an overlay of many SMFS recordings of the same category (e.g. high force unfolding pathway, low force unfolding pathway) on a single plot. This is quite common in the SMFS field and useful to guide the eye to quickly evaluate how robust and reproducible the measurements are.

To Whom It May Concern:

We thank the reviewers for the careful consideration, positive feed-back, and detailed comments on our work. Below, please find a point-by-point response to the reviewer comments (in gray), our response (in black), and changes in the revised manuscript (in green).

Reviewer #1:

Understanding the adhesion and mechanics of staphylococcal surface proteins is a hot topic with multiple medical and technical implications. In this interesting manuscript the authors unravel the mechanical stability of Ig-like folds (B domains) of staphylococcal adhesins, a longstanding challenge. The results are solid as well as the conclusions, no doubt the work will interest a broad audience.

1) last sentence of abstract: hinting at its role in for pathogen adhesion; I'm unclear how these key findings are that important for pathogenicity; the authors should qualify (remove the sentence) or elaborate further in conclusions.

The reviewer addresses an unclear point in the abstract. The role in pathogen adhesion mentioned refers to the “shock absorber” function introduced in the discussion. To avoid misunderstandings, we have simplified the last sentence to: “We show how calcium stabilizes an extremely strong protein fold of a pathogen adhesin in an unprecedented force regime.” p. 1

2) Intro, line 24. Worth to discuss Herman, mBio: In vivo, the B region of Cna is required for strong ligand binding and has been found to function as a spring capable of sustaining high forces of 1,2 nN. This previously undescribed mechanical response of the B region is of biological significance as it provides a means to project the A region away from the bacterial surface and to maintain bacterial adhesion under conditions of high forces.

The reviewer refers to previous work on a collagen binding homolog of SdrG, whose B region is stabilized by isopeptide containing B domains. Indeed, another very similar isopeptide containing homolog from *S. pyogenes* is mentioned in the discussion and shown in structural alignment to a B domain in the supplement (now Figure S9). These domains cannot completely unfold as their isopeptide bond forms a covalent lock between N- and C-terminus, which has been shown before (Echelman et al., PNAS 2016; Alegre-Cebollada et al. JBC 2010). A lack of complete domain unfolding will thus not permit work to be spent on it. Thus, the force response of such a system will largely be given by the entropic elasticity of the linkers that connect the domains and the overall adhesin to the bacterium (e.g. the long, unstructured and thus flexible SD-repeat region in SdrG). At high forces (> 1 nN) at which we use a modified FRC model as per Livadaru et al. (see methods) the force response of a peptide polymer for a given extension becomes almost linear, as can also be seen in our force extension curves in Figures 1e, 2a - resulting in the what may be described as a “spring” function.

The force spectroscopy data referred to by the reviewer (Herman-Bausier et al., mBio, 2016) on CNA was measured using a pulldown strategy via amines (i. e.

lysines). Pulldown via lysines will result in a large variety of pulling geometries, including multiple attachments as the authors specify, that each may result in different mechanical stabilities of the receptor-ligand investigated and differ from the mechanical strength in physiological, WT pulling geometry. These different approaches made us hesitant to compare the influence of CNA B domains to our site-specific pulldown-based experiments, which were conducted in the physiological pulling direction from the C-terminus of SdrG. We agree with the reviewer: the covalent isopeptide bond prevents the domain from unfolding, making a spacer function for CNA proteins very plausible. This concept could also be extended to the B region discussed in our work. We have included a reference to the publication, discussing the idea of a spacer function, now stating more explicitly: “The rigidity of B regions, especially those containing isopeptide bonds, such as the collagen adhesin Cna of *S. aureus*⁴⁰, has been proposed to project the ligand binding region away from the bacterial surface towards the host^{37,41,42}, which could also be a function of the B domains investigated here.” p. 6

3) Double peaks in fig 2a are reminiscent of the double peaks observed on live bacteria with SdrG (Herman, Mol Mic), worth to comment on this.

The citation in question is included in the manuscript, as this was the first *in vivo* mechanical probing of the SdrG system. Indeed, some of the curves in this work do contain final double peak events. The paper in question does not discuss B domain unfolding. As only direct distances, not contour length increment values, were provided for these data, we were hesitant to identify them as the B domains – although the forces and extension increments are consistent with our measurements. We have thus added a careful reference to the double peak observation – which could, as the reviewer suggested, have been a B domain unfolding event.

“Previous cell-based force spectroscopy work on SdrG had described an unfolding event preceding complex rupture at comparable forces and extension increments – yet not identified it as a B domain” p. 3

4) Ca plays a key role, but were other divalent tested?

The reviewer raises an interesting point, that the calcium binding loop could potentially be occupied by another divalent ion. Ion binding loops are usually very specific to a single ion species. We have tested if SdrG B1 can bind magnesium. At least at ~ mM magnesium concentrations no clear other unfolding pathway appeared. These results were mentioned in the submitted manuscript see page 4 line 6:

“After EDTA chelation, applying high concentrations of Mg²⁺ did not change SdrG B1 unfolding behavior, Mg²⁺ was unable to replace Ca²⁺ in the coordination sites.” A supplementary figure of this experiment has been added as new Figure S5, in which we have elaborated on these results in greater detail, the sentence now reads:

“After EDTA chelation, applying high concentrations of Mg²⁺ did not change SdrG B1 weak state unfolding behavior. At least at 18 mM, Mg²⁺ was unable to occupy the Ca²⁺ coordination sites (see Fig. S5).” p. 4

5) Conclusions. Comment on the link with the Cna paper above. Also a couple of staph adhesins show catch bond like behavior, e.g. the bond between ClfA and immobilized Fg (Herman Pnas). To what extent the role of Ca could be loading rate dependent and be linked to catch bond mechanisms?

We have speculated on a potential SdrG catch bond behavior in a previous publication, but have not published direct evidence of it as this requires a force ramp or clamp AFM SMFS experiment. Recent, unpublished data from our lab using a very slow force ramp experiment (force loading rates ~ 10 pN/s) suggest that SdrG:Fg β stays bound longer under force than would be permitted by its bulk off-rate - consistent with a catch bond behavior. However, these SdrG constructs do not contain B domains, thus we are confident in excluding B domains as a factor in this behavior.

As the B region in SdrG is not required for both mechanical stability of the ligand binding region (SdrG_N2N3:Fg β stability remains unchanged with or without the adjacent B-domains) and affinity (see Ponnuraj et al. Cell 2003) current data does not suggest that they have a direct influence in adhesin target binding and mechanical stability. These measurements do leave the scope of the present work which deals with B domain mechanics and were thus not included. We added this as an explicit statement to the discussion:

“In the case of SdrG the mechanical stability of the interaction between the N2 and N3 domains binding their Fg β target is independent of the B domains, as they can be deleted from the construct without lowering the interaction rupture force⁹.” p. 6

The role of B domains as spacers, as discussed for CNA, remains a possibility, we mention this more prominently now in the discussion with reference to the publication suggested by the reviewer (as mentioned above).

“The rigidity of B regions, especially those containing isopeptide bonds, such as the collagen adhesin Cna of *S. aureus*⁴⁰, has been proposed to project the ligand binding region away from the bacterial surface towards the host^{37,41,42}, which could also be a function of the B domains investigated here.” p. 6

Lastly, calcium does not seem to change the force loading rate behavior as we show in Fig. 2c. the slope of most probable unfolding force plotted against the log of the force loading rate of both the calcium saturated strong and calcium depleted weak state are almost the same, reflected in their very similar distance to the transition state Δx in the Bell-Evans model fit. We have stressed this point further in the main text and added the following to the results section:

“The dynamic force spectra for both weak and strong states, shown in Fig. 2c, were determined with a single cantilever. Notably, the dependency of the most probable rupture force on the natural logarithm of the force loading rate in the BE model is almost parallel for both states (strong state: $\Delta x = 0.083$ nm, $k_{\text{off}}^0 = 2.8\text{E}-17$ s⁻¹, weak state $\Delta x = 0.071$ nm, $k_{\text{off}}^0 = 0.011$ s⁻¹), reflected in similar distances to the transition state Δx , but a larger than 14 order of magnitude difference in zero force off-rate k_{off}^0 . ” p. 4

Reviewer #2:

Milles et al. present a very interesting paper in which they characterize how B domains stabilize by the coordination of Ca²⁺ ions. It is observed that the SdrD B1 and SdrG B2 domain folding is modulated from completely unfolding in EDTA chelating the Ca²⁺ ions to an extremely stable fold in higher Ca²⁺ concentrations. The possibility to modulate protein folding and stability by 'Ca²⁺ stimulation' is biotechnologically extremely interesting and promising. As such the authors outline that more complex structures involving B domains withstand extreme forces when contracted and folded in Ca²⁺, and upon exposure to Ca²⁺-chelating agents change into a flexible, extended polypeptide mesh. However, the exceptional mechanical and kinetic stability and calcium-tunable folding and force response suggest B domains as very promising protein design targets. I found the paper to be extremely well written, thought through and presented. It was a pleasure to read. The experiments are done at the highest standards (e.g., by leaders of the field) and the biological and biotechnological relevance of the paper is very high. I anticipate it will inspire various scientific fields and disciplines.

I have a few comments I would like to address to the authors

The authors write that the B2 domains may be used to sense Calcium. However. Calcium sensors of chemical and biological origin are well known and frequently used in for example live cell fluorescence imaging of calcium activities (gcamp etc). Possibly the authors could further line out why it may be needed to use a mechanical readout to sense calcium.

We agree with the reviewer, measuring the occupation of a weak or strong state by AFM-SMFS in a living cell seems impossible. The sensing aspect referred to would measure the presence of calcium by detecting the folding of the e.g. SdrD B1 domain by FRET dyes locate at N- and C-terminus of the domain, that would only exhibit FRET activity when the domain is folded, i.e. calcium present. We have clarified this ambiguous phrasing, stating now more explicitly:

“Furthermore, SdrG B2 and SdrD B1 may be used for Ca²⁺ sensing, as their folding upon Ca²⁺ binding could be read out by monitoring FRET of dyes attached at their N- and C-termini.” p. 7

The authors write that ‘It remains to be examined how B domains interact with each other, respond to constant forces or low force loading rates and what force the B domains exert when folding in the presence of Ca²⁺.’ Certainly, studying the domain stability depending on the listed parameters is interesting. However, to better understand the pathogenic function of the domains contributing to bacterial adhesion I would also suggest to characterize the B domain stability in dependence of other physiological parameters including pH, temperature, or other salts. Particularly the pH can change extremely from about 2-10.

We agree with the reviewer, the parameters space of B domain unfolding, and also refolding, was not been explored completely and we have added the suggested factors to the discussion, which now reads:

“It remains to be examined how B domains interact with each other⁴¹, respond to constant forces or low force loading rates, as well as changes in pH, temperature or ionic strength, and what force the B domains exert when folding in the presence of Ca²⁺. “ p. 7

The authors write about the physiological Calcium concentration. Please provide range and reference for your system.

The reviewer is correct, this value is missing. The physiological Calcium concentration in human blood is around 1 mM, far exceeding the dissociation constant of e.g. SdrG B1. We have added these values, including a reference, to the paragraph in question, it now reads:

“At physiological Ca²⁺ concentrations (free Ca²⁺ ions in human blood on the order of 1 mM³⁹) at least SdrG B1 would be found almost exclusively in its strong state. “ p. 6

Please define the temperature at which the experiments were performed.

The experiments were conducted at room temperature around 25 °C. We have added this piece of missing experimental conditions to the methods section:

“Experiments were conducted at room temperature (approximately 25 °C).” p. 15

Please provide errors of the spring constants determined.

The reviewer raises the point of force probe calibration confidence. Typically, the spring constant uncertainty in AFM based SMFS is in the vicinity of 10% (Brand et al., Meas. Sci. Technol. 2017). We have moved these values to the supplementary information with a reference to the work on spring constant uncertainty:

“The uncertainty of each value is approximately 10%⁵, making quantitative force comparisons between measurements challenging. When absolute comparisons were needed data were recorded with a single cantilever, e.g. in Fig. 3 c,d”
Supplementary methods

Reviewer #3:

The manuscript by Milles et al reports a fascinating atomic force spectroscopy study of the mechanical strength of beta domains of adhesins from pathogenic bacteria and its modulation by calcium. Prior to infection, bacteria such as *Staphylococcus aureus* adhere strongly to the host using highly specialized protein anchors that bind to receptors and need to withstand large pulling forces. In an earlier study, the same group, using AFM-based single molecule force spectroscopy (SMFS), determined that the molecular system mediating the adhesive interaction is unusually strong, with rupture forces exceeding 2000 pN (with a force of tens to a hundred pN being a typical strength of hydrogen bonds in biomolecular systems, for comparison). As immunoglobulin-like B domains are an integral part of the molecular system engaging in the adhesion (located upstream of the receptor-ligand part), the authors now asked the question how mechanically strong are these B domains. By attaching a single B domain at one end via a short linker to the tip of the AFM cantilever and the other end through N2N3 domains and the Fgbeta linker and some auxiliary reference modules to the glass substrate, they were able to directly pull on the B domain and measure its mechanical tensile strength. The SMFS results suggest that the B1 domain unfolds mechanically through two different pathways. The first pathways captured moderately large unfolding forces of around 600 pN and the second pathway reveals extremely high unfolding forces exceeding 2000 pN, suggesting a B1 domain with enormous mechanical stability. Moreover, the mechanical strength of this domain appears to be modulated by calcium ions in a concentration-dependent fashion, as calcium may occupy up to three binding sites within the domain. Mutations within these sites reveal that site # 3 is most important for the strength of the module. Similar SMFS measurements are then carried out for SdrG B2 domain from the same species and for the SdrD B1 domain from *Staph. aureus*. Essentially, all these B domains reveal extremely high mechanical stability with unfolding forces over 2 nN and also show more subtle differences in responding to calcium and in the proportion of “lower mechanical stability pathway”. Overall the work is carried out with great attention to detail, particularly with respect to generating various constructs and mutants and carrying out a significant number of measurements at different loading rates.

However, I have three methodological observations and concerns that I wish the authors consider:

a) Including the ddFLN4 domain in the construct is clearly beneficial for identifying single molecule AFM recordings, as this domain provides an unambiguous mechanical unfolding fingerprint in force-extension curves. However, by itself, a construct which has the ddFLN4 domain only on one side of the B domain of interest (as used by the authors) cannot provide undisputable evidence that the B domain was subjected to stretching forces and unfolded completely in a given experiment. The contour length increment following the first large unfolding peak being consistent with the length of the B domain helps to suggest that this unfolding peak originated from the B domain, but by itself, does

not provide direct evidence. For this, one needs to have the ddFLN4 domain (or another fingerprint generating domain) also on the other side of the B domain in the protein construct.

The reviewer raises the point of confidence in the unambiguity of the unfolding increment assigned to the B domain. In previous work we were able to establish that no increment matching the B domain was present in force curve in which SdrG_N2N3 (N2N3 being the Fg β peptide binding domains) was tethered with Fg β -ddFLN4. The ddFLN4 increment was visible exclusively (Milles et al., Science, 2018). When the SdrG B1 domain was included in the protein, now using the same Fg β -ddFLN4-ybbr pulling SdrG_N2N3-B1-ybbr the additional ~ 36 nm contour length increment appeared, changing its unfolding force depending on the presence of EDTA or calcium. We take this, and as the reviewer mentioned, the contour length increment matching the expected value for it as sufficient evidence to assign this increment to the B domain. The suggestion to add another fingerprint onto the system, effectively sandwiching the B domain between two other domains, would be a required approach when using a nonspecific polyprotein tethering protocol – in which one needs to establish that the force did propagate through the domain of interest, here the B1 domain. As we used site specific immobilization with the ybbr tags at the C-termini of each construct and established the SdrG_N2N3:Fg β tethering beforehand in our previous work, force must propagate through the B1 domain and we can assign this increment with very high confidence to its unfolding. The added contour length diagrams and representative traces, added as per point b) of the reviewer, in the new Figures S2 and S3 should clarify this point.

b) As mentioned above, contour length increments following the first unfolding peak attributed to the B domain are very important but the data about distributions (pdf) of this length increment is not included in the manuscript. I am sure many readers would like to see those distributions (for all B domains and under various calcium conditions) to be shown directly as accompanying unfolding force distributions as they are equally important to this study.

The reviewer raises the point of contour length diagrams data being directly presented. We concur, this data has now been included in the supplement as Figure S3. The graphs clearly show the increment for B domain unfolding staying constant in both the strong and weak unfolding pathways. This should also help the address the reviewers point concerning the B1 contour length increment raised above. The changes now read:

“When probed in 10 mM EDTA, the stability of SdrG B1 dramatically decreased, and the previously described “weak” unfolding event ~ 600 pN appeared exclusively (a set of representative force-extension curves is shown in Fig. S2, for contour length diagram alignments see Fig. S3).” p.3

c) It would be useful to show, either in the main manuscript or in supporting information, an overlay of many SMFS recordings of the same category (e.g. high force unfolding pathway, low force unfolding pathway) on a single plot. This is quite common in the SMFS field and useful to guide the eye to quickly evaluate how robust and reproducible the measurements are.

We agree with the reviewer's suggestion. A set of representative force-extension curves for both strong and weak unfolding pathway of SdrG B1 has been added to the supplement, Figure S2. This is referenced as above:

“When probed in 10 mM EDTA, the stability of SdrG B1 dramatically decreased, and the previously described “weak” unfolding event ~ 600 pN appeared exclusively (a set of representative force-extension curves is shown in Fig. S2, for contour length diagram alignments see Fig. S3).” p.3

Reviewers' Comments:

Reviewer #1:

Remarks to the Author:

I am very happy with revisions raised by all referees.

Reviewer #3:

Remarks to the Author:

The revised manuscript now includes (in Supplementary Information) multiple examples of force extension curves for SdrG B1 under various calcium conditions, and they are very consistent. Similarly, including now in the revised manuscript probability density distributions versus relative contour length, strengthens the work. Thus, I am completely satisfied with the authors' responses and revisions regarding my concerns b) and c) (the original review).

Regarding point a) about my suggestion to include an additional protein between the B1 domain and the AFM tip, I agree with the authors' response partially. Clearly, specific (versus nonspecific) immobilization of the construct to the AFM tip is very helpful in making sure that the applied force propagates through the B1 domain. However, the B1 domain remains fairly close to the AFM tip in those measurements, so unwanted interactions between this domain and the tip cannot be totally excluded. Thus, in my opinion having another fingerprint would help to alleviate any concerns that the unusually high force measured by the AFM may be due to these nonspecific unwanted interactions (which generally can be very strong). This suggestion was aimed at providing an additional layer of evidence that would even further strengthen the observations in this work about extremely strong mechanical stabilities of B domains. But since the authors being pioneers in the field, feel confident about their approach as it is, I do not insist on this point.